# Different Forms of Disorder in NMDA-Sensitive Glutamate Receptor Cytoplasmic Domains Are Associated with Differences in Condensate Formation

**DOI:** 10.3390/biom13010004

**Published:** 2022-12-20

**Authors:** Sujit Basak, Nabanita Saikia, David Kwun, Ucheor B. Choi, Feng Ding, Mark E. Bowen

**Affiliations:** 1Department of Physiology and Biophysics, Stony Brook University, Stony Brook, NY 11794, USA; 2Department of Chemistry, Navajo Technical University, Crownpoint, NM 87313, USA; 3Quantum-Si, Inc., Guilford, CT 06437, USA; 4Department of Physics and Astronomy, Clemson University, Clemson, SC 29634-0978, USA

**Keywords:** glutamate receptor, intrinsically disordered protein, discrete molecular dynamics, single molecule fluorescence, liquid-liquid phase separation

## Abstract

The N-methyl-D-aspartate (NMDA)-sensitive glutamate receptor (NMDAR) helps assemble downstream signaling pathways through protein interactions within the postsynaptic density (PSD), which are mediated by its intracellular C-terminal domain (CTD). The most abundant NMDAR subunits in the brain are GluN2A and GluN2B, which are associated with a developmental switch in NMDAR composition. Previously, we used single molecule fluorescence resonance energy transfer (smFRET) to show that the GluN2B CTD contained an intrinsically disordered region with slow, hop-like conformational dynamics. The CTD from GluN2B also undergoes liquid–liquid phase separation (LLPS) with synaptic proteins. Here, we extend these observations to the GluN2A CTD. Sequence analysis showed that both subunits contain a form of intrinsic disorder classified as weak polyampholytes. However, only GluN2B contained matched patterning of arginine and aromatic residues, which are linked to LLPS. To examine the conformational distribution, we used discrete molecular dynamics (DMD), which revealed that GluN2A favors extended disordered states containing secondary structures while GluN2B favors disordered globular states. In contrast to GluN2B, smFRET measurements found that GluN2A lacked slow conformational dynamics. Thus, simulation and experiments found differences in the form of disorder. To understand how this affects protein interactions, we compared the ability of these two NMDAR isoforms to undergo LLPS. We found that GluN2B readily formed condensates with PSD-95 and SynGAP, while GluN2A failed to support LLPS and instead showed a propensity for colloidal aggregation. That GluN2A fails to support this same condensate formation suggests a developmental switch in LLPS propensity.

## 1. Introduction

The N-methyl–D-Aspartate (NMDA)-sensitive glutamate receptor (NMDAR) plays a pivotal role in excitatory synaptic transmission and synaptic plasticity, which impacts learning, memory, and cognition [1,2]. NMDARs are heterotetrametric formed from two GluN1 and two GluN2 subunits, which can be GluN2A, GluN2B, or mixtures of different isoforms [3,4]. NMDARs have four structurally-separable domains: the extracellular amino terminal domain (ATD), the ligand binding domain (LBD), the transmembrane domain (TMD), and the intracellular C-terminal domain (CTD). These domains work together, enabling NMDARs to function as a ligand-gated ion channel. The binding of glutamate and glycine to the extracellular LBDs propagates a conformational change leading to the opening (i.e., gating) of the ion conduction pore in the transmembrane domain [3,5]. The gating propensity is further modulated by both the extracellular ATD [6] as well as the intracellular CTD [7,8].

Whole-exome sequencing revealed that mutations in NMDARs are associated with neuropsychiatric disorders [9]. While the majority of mutations are found within the LBD and TMD, several disease-associated mutations fall within the CTDs of GluN2A and GluN2B [10,11]. Knowledge of NMDAR structure is necessary to understand the molecular basis of these disorders. The structure of NMDARs is almost entirely known, from structural studies of the extracellular and transmembrane domains [12,13]. Thus, our understanding of the mechanism for ligand-induced gating is nearly complete [4]. However, structural information about the intracellular CTD has proved elusive due the presence of intrinsic disorder [14,15,16]. In GluN2A and GluN2B, the CTD is the largest single domain in the protein and appears to be split into two separate subdomains (CTD1 and CTD2) by a central palmitoylation motif [17] (Figure 1A). The full CTDs have never been characterized due to their limited solubility. Previously, we confirmed experimentally the presence of intrinsic disorder in CTD2 from GluN2B and identified slow timescale conformational dynamics [18]. However, no information is available for GluN2A.

In addition to allosteric modulation of gating, the CTD plays a major role in the formation of postsynaptic signaling complexes through interactions with the scaffolding protein post synaptic density protein of 95 kDa (PSD-95) [21] along with numerous other signaling proteins [22]. Thus, the CTD plays a role in the initiation of signaling cascades, which is separate from its role in ion channel gating [23,24]. Recent reports have shown that PSD-95 and the GluN2B CTD are capable of liquid–liquid phase separation (LLPS) in vitro with a recombinant synGAP [25,26,27,28]. Proteins containing intrinsic disorder are key players in LLPS because exposed aromatic “sticker” residues enable multivalent interactions [29,30,31]. The postsynapse has long been known to contain condensates, which have been termed the postsynaptic density (PSD) [32,33,34]. The formation of condensates in both the presynapse and postsynapse have been linked to LLPS [35,36].

Here, we compared the CTD2 domains from GluN2A and GluN2B using sequence analysis, discrete molecular dynamics simulations (DMD), and single molecule FRET (smFRET). Analysis of the amino acid sequences suggested differences between the subunits in the form of disorder [20]. DMD revealed differences in polypeptide compaction, with GluN2A favoring extended states while GluN2B remained globular. We did not observe any slow timescale dynamics in single molecule fluorescence measurements GluN2A, which we previously observed in GluN2B [37]. To understand how these differences in disorder affected protein interactions, we compared GluN2A and 2B for the ability to undergo LLPS using sedimentation and differential interference contrast (DIC) microscopy. This revealed that GluN2A was not capable of supporting LLPS while GluN2B lowers the concentration regime for phase separation with PSD-95 and synGAP [25]. Given the developmental switch in these receptor isoforms [38,39], this would imply an associated switch in LLPS propensity at the synaptic membrane with a higher propensity for LLPS during early development and then decreasing LLPS propensity as GluN2A comes to predominate.

## 2. Materials and Methods

### 2.1. Protein Purification

The C-terminal domain 2 (CTD2) of GluN2A (residues 1239–1464, CTD2A) and of GluN2B (residues 1259–1482, CTD2B) from *Rattus norvegicus* were expressed in the Rosetta strain of *Escherichia coli* (MilliporeSigma, Burlington, MA, USA) from the expression vector pPROEX HTB (ThermoFisher Scientific, Waltham, MA, USA), which imparts an N-teminal 6-His tag [15,40]. The CTD2 cell pellets were lysed under denaturing and reducing conditions, which were maintained during affinity purification. For CTD2B, the protein was eluted in denaturant free buffer [40], but for GluN2A, the protein was maintained in a nondenaturing concentration of urea (2 M) to prevent aggregation. The 6-His tags were removed using tobacco etch virus protease (TEV), which is unaffected by 2M urea. Subsequent rounds of cation exchange and size exclusion chromatography on Superdex S-200 (Cytiva, Marlborough, MA, USA) were used to obtain protein purity of 95% or greater as verified using SDS-PAGE. Full-length PSD-95 from *Rattus norvegicus* was expressed in the Rosetta 2 strain of *E. coli* and purified by a combination of Ni-affinity, anion exchange, and size exclusion chromatography as previously described [41]. The recombinant construct containing the N-terminal coil-coiled (CC) fused to the PSD-95 binding motif (PBM) of synGAP was a kind gift from Mingjie Zhang and was expressed and purified as described [25].

To enable fluorescent labeling, we used two native cysteines in CTD2A (C1239 and C1412) with the three remaining native cysteines (C1241, C1387, and C1448) changed to serine through classic site-directed mutagenesis as confirmed by DNA sequencing. For CTD2B, there was not a suitable native cysteine pair, so we introduced a cysteine at S1273 and paired this with a native cysteine at 1445. The two remaining native cysteines in CTD2B (C1394 and C1455) were mutated to serine.

### 2.2. Single Molecule Total Internal Reflection Fluorescence (smTIRF) Microscopy

The purified CTD2s were randomly labeled with an equimolar ratio of Alexa Fluor 555 C5 maleimide and Alexa Fluor 647 C2 maleimide (Thermo Fisher Scientific) overnight at 4 °C in 25 mM HEPES, pH 7.4, 300 mM NaCl, and 0.5 mM tris (2-carboxyethyl) phosphine (TCEP). All buffers included 2M urea for CTD2A. Unconjugated dye was removed by desalting with Sephadex G50 (Cytiva, Marlborough, MA, USA) followed by dialysis. Fluorescently labeled CTD2s were N-terminally biotinylated by adding a 5-fold molar excess NHS-LC-Biotin with a ~2 nm spacer (ThermoFisher Scientific, Waltham, MA, USA) in 50 mM potassium phosphate buffer at pH 6.5 to direct the reaction to the N-terminus. The reaction mixture was incubated at 4 °C overnight, followed by desalting to remove free biotin.

Biotinylated proteins were attached via streptavidin to a quartz slide passivated with biotinylated BSA and a mixture of Biolipidure 203 and 206 (NOF AMERICA Corporation, White Plains, NY, USA). Alternating illumination using diode lasers at 532 nm (Laser Quantum) and 640 nm (Coherent) allowed for the identification of optically resolved single molecules containing one donor and one acceptor. Samples were excited using prism-based TIRF. Images were acquired on an Olympus IX-71 microscope with a 60X-1.2 NA water-immersion objective. Fluorescence emission collected from donor and acceptor were spectrally separated using an optosplit emission image splitter (Cairn Research, Faversham, UK) and relayed onto a single Andor iXon EMCCD camera (Andor Technology, Ltd., Belfast, UK). Data were collected at 10 frames/second. All smFRET measurements were performed in 50 mm tris, 200 mM NaCl, pH 7.4, and supplemented with 1 mM cyclooctatetraene, 0.8% *w/v* glucose, 7.5 units/mL glucose oxidase, and 1000 units/mL catalase. Microscopy data were analyzed using MATLAB to correlate donor and acceptor images, extract single molecule intensity time traces, and calculate FRET efficiency [42].

### 2.3. Discrete Molecular Dynamics (DMD) Simulations

DMD is a molecular dynamics algorithm that has been shown to have high predictive power and sampling efficiency in studying conformational dynamics of IDPs [43,44,45]. Details of DMD methods can be found in [46,47]. To sample the conformational free energy landscape efficiently, we performed replica exchange DMD simulations with 18 neighboring replicas in the temperature range of 275–360 K. Both proteins started from an extended conformation and reached equilibrium quickly in DMD simulations as indicated by the distributions of the radius of gyration and secondary structure contents (Figure 2). We used the conformations sampled in rxDMD within the temperature range of 300–310 K to compare the conformational difference between CTD2A and CTD2B.

The secondary structure was calculated using the DSSP program. The hydrogen bond was considered to be formed when the N⋯O distance was within 3.5 Å, and the N–H⋯O angle was more than 120°. A pairwise residue contact was defined as the distance between the heavy atoms from the main chain or side chain of two nonsequential residues within 0.65 nm.

### 2.4. Measurement of Turbidity 

Full-length PSD-95, CC-PBM from SynGAP, and CTD2A or CTD2B were mixed at 1:1:1 ratio in 20 mM tris, 150 mM NaCl, at pH 7.4 buffer with each protein at a final concentration of 20 µM. Samples were equilibrated in polypropylene microfuge tubes at room temperature (25 °C) for 5 min before measurements. Transmittance of aliquots removed from the incubation was measured at 550 nm in quartz cuvettes with a 1 cm path length using an Agilent model 8453 UV-Vis spectrophotometer. Turbidity was calculated as percentage of transmittance. The time-dependent turbidity formation seen with CTD2A was slower than condensate formation induced by the ternary mixture. As such, we can remove urea from CTD2A for condensate formation experiments without the appearance of CTD2A precipitation.

### 2.5. Sedimentation Analysis of Condensates

Full-length PSD-95, SynGAP CC-PBM, and CTD2A or CTD2B were mixed at 1:1:1 ratio in 20 mM tris-HCl, 150 mM NaCl, at pH 7.4 buffer with each protein at a final concentration of 20 µM. Samples were equilibrated in polypropylene microfuge tubes at room temperature (25 °C) for 5 min before sedimentation at 13,200× *g* for 1 min. The isolated pellets were suspended in the original volume of buffer. Then, supernatant and pellet fractions were boiled in Laemmli buffer containing dithiothreitol and resolved by SDS-PAGE on 4–20% gradient gels, which were stained with Coomassie Brilliant Blue R-250 (Bio-Rad).

### 2.6. Differential Interference Contrast (DIC) and Fluorescence Microscopy

Full-length PSD-95, SynGAP CC-PBM, and CTD2A or CTD2B were mixed at 1:1:1 ratio in 20 mM tris-HCl, 150mM NaCl, at pH 7.4 with each protein at a final concentration of 20 µM. Samples were equilibrated in 8 chamber slides (Nunc, Lab-TEK II) at room temperature (25 °C) for 5 min before imaging. The chamber was passivated with BSA to avoid nonspecific interactions with the coverslip. The samples were imaged on a Nikon Eclipse Ti-E microscope with a 100X 1.4 NA oil-immersion objective equipped with prisms for DIC imaging and a Nikon total internal reflection fluorescence (TIRF) excitation module connected to a fiber-coupled laser launch. Images were recorded with an iXon electron multiplying charge-coupled device camera (Andor Technology, Ltd., Belfast, UK) and analyzed using Nikon Elements for background subtraction.

Proteins containing two unique cysteine residues (taken from our previous work [15,41]) were expressed and purified as described for wild type proteins. Full-length PSD-95, containing the mutations S398C and R492C, was labeled with Alexa 488 maleimide. CTD2B, containing the mutations S1273C and C1445, was labeled with Alexa 647 maleimide. The labeled proteins were isolated from the free dye by desalting with Sephadex G-50. The labeling efficiency was >98% for both proteins as determined with absorbance spectroscopy using the calculated extinction coefficients.

For imaging, 300 nM of labeled protein was used along with full-length PSD-95, CC-PBM from SynGAP, and CTD2B at 1:1:1 ratio in 20 mM Tris-HCl, 150 mM NaCl, at pH 7.4 at 20 µM concentration. Samples were mixed in 8 chamber slides (Nunc, Lab-TEK II) at room temperature (25 °C) for 5 min before imaging. The chamber was passivated with BSA to avoid nonspecific interactions with the surface. Laser excitation was introduced using highly inclined and laminated optical sheet (HILO) microscopy [48]. The samples were imaged with visible light using a DIC prism, HILO at 488 nm, and HILO at 642 nm. Fluorescence emission was separated from laser excitation using a 405/488/561/642 multiband filter set (Chroma Technology Corp). Images were recorded with an iXon electron multiplying charge-coupled device camera (Andor Technology, Ltd.) and analyzed using Nikon Elements for background subtraction.

## 3. Results

### 3.1. Primary Sequence Analysis

The GluN2A and GluN2B subunits from *Rattus norvegicus* share a 71% sequence identity throughout their ordered extracellular and transmembrane domains. However, the sequence conservation drops to only 31% sequence identity within the CTDs. The sequence similarity is higher at 47% because the overall chemical composition is similar with a high proportion of serine and asparagine. Both CTDs contain two conserved cysteine clusters, which have been shown to be sites of palmitoylation that lead to membrane attachment once post-translationally modified [49,50]. Thus, the CTDs from both GluN2A and GluN2B share this organization of two subdomains demarcated by internal palmitoylation clusters, which we have termed CTD1 and CTD2 (Figure 1A). Excluding the palmitoylation motifs, the 26% sequence identity within CTD1 is slightly lower than within CTD2 at 36% identity (Table 1).

Previously, we used PONDR to show that the CTD from GluN2B was predicted to contain intrinsically disordered regions (IDRs) [18,52,53]. Here, we used PONDR to compare the distribution of IDRs within GluN2A and GluN2B (Figure 1B). From this analysis, we observed that both subunits have similar predictions of an order-forming region after the transmembrane domain, which is broken up by an IDR. In GluN2B, CTD1 is predicted to be order-prone to around residue 1075 with only short disordered motifs. In contrast, the GluN2A CTD1 is predicted to contain a long IDR between residues 915 and 987. Both isoforms also have a prediction of an IDR at the beginning of CTD2, which is longer in GluN2B. However, the distal half of CTD2 in GluN2B is predicted to be order-prone until just before the C-terminus. In contrast, the distal half of CTD2 in GluN2A contains a mixture of short disorder and order-prone motifs. The C-terminus of both isoforms contains the PSD-95 binding motif. The two isoforms differ with GluN2A containing a predicted IDR preceding the C-terminal PSD-95 binding motif, while GluN2B is predicted to be order-prone.

To provide more detail on the differences in sequence features between these isoforms, we performed Classification of Intrinsically Disordered Ensemble Regions (CIDER) for the CTDs [19]. Interestingly, the CTDs from both GluN2A and GluN2B have a relatively low fraction of charged residues (FCR) and low net charge per residue (NCPR) for a protein containing intrinsic disorder (Table 1), which is often associated with a high FCR [51,54,55]. In CTD1, the FCR was comparable for both isoforms and the low NCPR values classify them as weak polyampholytes (Figure 1C). The segregation of positive and negative charges within the polypeptide (kappa [20]) was 28% higher in the GluN2B CTD1, although both isoforms were relatively well-mixed. The CTD2 from GluN2A showed a 27% higher FCR than GluN2B (0.206 and 0.261, respectively). Surprisingly, the GluN2B CTD2 showed an almost 3-fold higher NCPR than GluN2A (NCPR = 0.026 and 0.009, respectively). GluN2B also had a slightly higher kappa value in both subdomains, indicating a higher degree of charge segregation. Overall, both CTD2s have fewer charged residues compared to CTD1, but the charges are more segregated in CTD2, which can influence the form of disorder [20,31].

According to our CIDER analysis, both CTDs are classified as disordered globules rather than extended polymers [20]. Both CTD1 and CTD2 from GluN2A, along with CTD1 of GluN2B, lie on the border between strong and weak polyampholytes, which makes their conformational behavior hard to predict. In contrast, the CTD2 of GluN2B is classified as a weak polyampholyte (Figure 1C, inset). Protein sequences in this region of the CIDER plot have a high tendency to form collapsed globules [56]. Based on this analysis, the amino acid sequence of CTD2 from GluN2B appears to have evolved to adopt a different form of intrinsic disorder.

Recombinant constructs based on the CTD2 subdomain of GluN2B have been shown capable of participating in LLPS [25,26,27,28]. LLPS in IDPs has been linked to amino acid patterning, particularly of aromatic and arginine residues, which participate in cation–π and π–π interactions [29,31]. Similarly, the distribution of charged residues has been linked to the form of intrinsic disorder [20]. To analyze residue patterning within CTD2, we plotted the separation between repeated amino acids in boxplot format along with the Gaussian distribution of their frequency (Figure 1D). Both GluN2A and GluN2B have a similar number of aromatic residues within CTD2 with similar frequency. In GluN2A, these tend to be tyrosine, whereas in GluN2B, phenylalanine predominates. In GluN2A, the frequency of arginine residues is half that of the aromatic residues (10 ± 10 compared to 21 ± 14, respectively; *p* = 0.015), while in GluN2B, the frequency of arginine and aromatic residues is the same (19 ± 17 compared to 22 ± 18, respectively). Additionally, GluN2A has a higher density of negatively charged residues along with fewer lysines, resulting in only four unpaired arginine residues. In contrast, GluN2B has fewer negatively charged residues along with more lysines, which results in eight unpaired arginine residues. Thus, the GluN2B CTD2 has matched arginine and aromatic residue patterning that appears favorable for the cation–π interactions, which support LLPS, while GluN2A appears to be dominated by electrostatic interactions resulting in the low NCPR.

### 3.2. Discrete Molecular Dynamics

Based on amino acid sequence analysis, the CTD2 domains were predicted to adopt different forms of intrinsic disorder (Figure 1C). To understand how this difference manifests in the conformational free energy landscapes, we used replica-exchange discrete molecular dynamics (rxDMD) with 18 replicas, running at different temperatures, for a combined simulation time of 8.0 µs. The predictive power of DMD with the enhanced sampling of replica-exchange is well suited to describing the energy landscape of IDPs and folded proteins [43,44,45,57]. We performed rxDMD simulations for the CTD2 subdomain from both GluN2A and GluN2B as free polypeptides (Figure 2). Both proteins showed a highly dynamic and variable conformation. Examination of the radius of gyration (R_g_) for the individual conformations sampled during the rxDMD trajectory revealed that GluN2A favored extended conformations starting at 40 Å but extending to 100 Å (Figure 2A). In contrast, GluN2B favored compact states with mode radii around 30 Å. However, the GluN2B R_g_ distribution did contain a second peak with extended states out to 60 Å (Figure 2B). By analyzing all the snapshots from the DMD trajectory, we could calculate the secondary structural propensity along the polypeptide chain, which agreed well with PONDR predictions. Both CTD2 started with a pair of short α-helices, followed by a disordered region, which is periodically interrupted by structured elements in GluN2A but continues uninterrupted in GluN2B (Figure 2C,D). The low propensity for secondary structure in GluN2B is in good agreement with our previous circular dichroism measurements [40].

Examination of the pairwise contact maps from rxDMD revealed few persistent long-range interactions in either CTD2s, as expected for IDPs (Figure 2E,F). However, comparison of the contact frequency maps revealed differences in medium- and short-range contacts (~20 to 60 residue separation), which were less pronounced in GluN2B. In contrast, GluN2A showed a central region with persistent contacts suggesting an order-prone domain. Additionally, the pairwise contact map shows the strongest short-range contacts in GluN2A at the C-terminus (Figure 2E). Thus, rxDMD found that both CTD2s share a helical region following the palmitoylation motif but diverge after this point. In GluN2A, there is a mixture of secondary structural elements along with two regions showing persistent contacts, which is in good general agreement with PODR predictions. In contrast, GluN2B was largely disordered throughout its length with minimal persistent contacts.

### 3.3. Single Molecule Fluorescence Resonance Energy Transfer (smFRET)

We previously used smFRET to show that recombinant CTD2 from GluN2B (CTD2B) displayed slow timescale conformational dynamics, which we termed hop-like intramolecular diffusion [15,18,37,40]. To probe conformational dynamics in GluN2A with smFRET, we created a recombinant CTD2 from GluN2A (CTD2A), which retained two native cysteines (C1239-C1412) for a separation of 173 residues. For CTD2B, there were no native cysteines with similar separation, so we paired the S1273C mutation with the native C1445 to achieve a separation of 172 residues. Thus, the contour length of the polypeptide between the points of measurement are similar for both protein constructs.

We immediately noticed differences between the CTD2 constructs during recombinant expression. While the CTD2B is highly soluble, CTD2A was prone to self-association displaying a slow accumulation of colloidal turbidity over time that eventually led to precipitation. We found that inclusion of urea was sufficient to forestall this process during protein handling and could be removed before any measurements. Proteins were randomly labeled to completion with an equimolar mixture of the Alexa 555 donor and the Alexa 647 acceptor dyes. The labeled protein was then selectively biotinylated at the N-terminus and attached to a passivated microscope slide that was functionalized with streptavidin. Once the proteins were surface attached, we removed all urea by rinsing and made measurements under urea-free conditions. The optical resolution between molecules allowed no possibility of intramolecular aggregation. Samples were excited using prism-based total internal reflection with alternating laser excitation to identify single molecules containing an active donor-acceptor pair.

Examination of the individual time traces for CTD2A revealed steady intensity until photobleaching but individual molecules persisted in high, medium, or low FRET states (Figure 3A). This is in stark contrast to the stochastic intensity transitions that we have repeatedly observed for CTD2B (Figure 3B) [18,40]. When we accumulated the molecules into population histograms, we observed that CTD2A showed three well-resolved peaks in the distribution: a low FRET peak encompassing 24% of the population along with a broader peak at intermediate FRET with 24% occupancy and a predominant high FRET peak with 52% occupancy (Figure 3C). In contrast, the population histogram for CTD2B showed a wide distribution that was fit by two broad peaks at low FRET and intermediate FRET (20% and 80% occupancy, respectively) without a distinct peak at high FRET (Figure 3D). The low and intermediate FRET peaks were of similar efficiency in both isoforms but much narrower in CTD2A than CTD2B, suggesting differences in the rate of conformational exchange [58,59]. The conformational dynamics of these IDPs are orders of magnitude faster than the time resolution of data collection (10 Hz). As such, the histograms represent the time-averaged distribution of states and do not provide information about the underlying rapid dynamics.

Surprisingly, we observed a higher FRET state in CTD2A, suggesting a collapsed state, which was not observed in CTD2B. Such collapsed states were not observed in rxDMD simulations of CTD2A. However, CTD2A was directionally attached via the N-terminus to a passivated surface for measurements, which mimics the membrane attachment that occurs upon palmitoylation, while rxDMD simulations were of free protein. We previously showed that CTD2B favored more condensed states when directionally attached to a surface relative to the conformation in solution [15].

### 3.4. Condensate Formation

Previously, recombinant constructs based on CTD2 from GluN2B have been shown to undergo LLPS in vitro with the synaptic scaffold PSD-95 and a redesigned construct based on synGAP that fuses the Coiled-Coil domain to the PSD-95 Binding Motif (CC-PBM) [26,28]. However, condensate formation has not been examined with GluN2A. We examined condensate formation by monitoring transmittance at 550 nm to measure the turbidity of protein mixtures. The formation of condensates is highly sensitive to solvent conditions [61], so we performed all experiments at room temperature (25 °C) in tris-buffered saline (20 mM tris 150 mM NaCl pH 7.4). All the individual proteins showed 100% transmittance at 20 µM, which indicates a lack of condensate formation. Among all the binary protein combinations, only PSD-95 with GluN2B CTD2B showed any turbidity as a binary mixture with transmittance at 36%, which agrees well with previous binary LLPS experiments [28]. As expected, the ternary solution of PSD-95, CC-PBM, and CTD2B showed a very low transmittance of 5% indicating robust condensate formation (Figure 4A). In contrast, CTD2A showed no signs of turbidity under the exact same conditions.

To examine the protein composition of the condensates, the ternary mixtures containing PSD-95 and CC-PBM with CTD2A or CTD2B were centrifuged to separate the condensed phase from the dilute phase [62]. The sedimented pellets were dissolved in the same volume as the original supernatant and then resolved with SDS-PAGE to examine the partitioning of individual proteins into condensates. As expected from turbidity measurements, there was no protein pellet for CTD2A (Figure 4B), whereas the condensates isolated using CTD2B contained both PSD-95 and CC-PBM. To provide further evidence, we examined the ternary solution of a 1:1:1 ratio containing PSD-95, CC-PBM, and CTD2 using differential interference contrast (DIC) microscopy. We observed that 20 µM CTD2A remained clear (Figure 4C). In contrast, the ternary mixture with 20 µM CTD2B formed a dispersion of spherical droplets with a range of diameters (Figure 4D). To provide additional confirmation that CTD2B was located within the droplets, we used cysteine variants from our previous work [15,41] to label PSD-95 Alexa 488 and label CTD2B with Alexa 647. We performed two-color imaging by including 300 nM of each labeled protein to the ternary solution of a 1:1:1 ratio PSD-95, CC-PBM, and CTD2. Both labeled proteins localized to the same droplet. Thus, all droplets visible by DIC contained CTD2B and PSD-95 (Appendix A) in agreement with our SDS-PAGE analysis (Figure 4B).

## 4. Discussion

The NMDA receptor is an obligate heterotetramer containing two GluN1 subunits and two GluN2 subunits, which are predominantly GluN2A or GluN2B in the cortex and hippocampus [3,4]. The ordered extracellular and transmembrane domains in NMDARs form a ligand–gated ion channel. Despite high sequence conservation in these domains, receptors containing only GluN2A are functionally distinct from those containing only GluN2B, both in terms of their channel properties [63] and also in their downstream signal transduction [22]. The most variable domain in NMDAR subunits is the intracellular CTD, which has evolved to be the largest domain in GluN2A and GluN2B [16,64]. Despite the low sequence conservation in their CTDs, these two isoforms share a similar arrangement of two “domains” demarcated by palmitoylation sites [17]. Whether these are truly domains in the structural sense remains unclear.

The CTD subdomains from GluN2A and GluN2B share little sequence homology with each other or other known proteins. The CTDs are predicted to contain a mixture of order-forming and disordered motifs (Figure 1B). The entire GluN2A CTD and CTD1 from GluN2B share a similar amino acid composition with a low net charge on the boundary of strong and weak polyampholytes (Figure 1C), which makes their conformational behavior hard to predict. In contrast, CTD2 from GluN2B was classified as a weak polyampholyte, mostly due to differences in amino acid patterning, which favors collapsed states. Weak polyampholytes form globule or tadpole-like conformations while strong polyampholytes can form coil-like conformations or admixtures [20].

While simulations have been used to understand ligand binding and gating in NMDARs [65,66,67,68], we present the first simulations involving the CTD. In agreement with our CIDER classification of CTD2 from GluN2A and GluN2B into different conformational classes, our rxDMD simulations revealed large differences in polypeptide extension and secondary structural propensity. PONDR prediction of GluN2A showed a mixture of ordered and strongly disordered motifs, which agrees well with the interspersion of α-helical and β-sheet conformation within a framework of random coil. These local structural elements give rise to strong short-range interactions in the contact frequency map, particularly around the PSD-95 binding motif (Figure 2E). The presence of local structured elements in GluN2A has the effect of increasing the net polypeptide expansion by preventing a globular collapse, which is largely what we observed in GluN2B. There were almost no persistent intramolecular contacts in GluN2B (Figure 2E). Thus, rxDMD observed a collapsed globule with almost no secondary structure that remained highly dynamic. This seems at odds with the PONDR prediction of an order-prone domain within the GluN2B CTD2 (Figure 1B). However, GluN2B had a much smaller R_g_ suggesting that the PONDR prediction may be identifying the propensity to undergo globular collapse rather than becoming ordered through persistent contacts.

To date, only recombinant constructs based on CTD2 from GluN2B have been characterized experimentally. Here, we present experimental characterization of CTD2 from GluN2A. We found that CTD2A was poorly soluble compared to CTD2B. The slow accumulation of turbidity in CTD2A during protein handling was prevented with urea that was removed before any measurements. Using camera detection to measure smFRET, the fast conformational dynamics were time-averaged, which would result in a single time-averaged peak for random coil-like IDPs [58,59]. However, we saw three distinct, narrow peaks in the population histogram for CTD2A. Single molecules showed a stable energy transfer until photobleaching, suggesting a static heterogeneity across the population. This is in contrast to CTD2B, which showed two broad peaks with dynamic, anticorrelated intensity transitions at the single molecule level (Figure 3B,D). Thus, CTD2A lacks the slow timescale stochastic transitions seen in CTD2B (and other IDPs) using smFRET [37].

We are hesitant to interpret the changes in energy transfer in terms of distance given the dynamic environment of the fluorophores. However, simple calculations based on a self-avoiding walk (SAW) polymer model suggest similar polypeptide extension for the low and intermediate FRET states in both isoforms (Table 2). We also observed a high FRET peak in CTD2A, which suggests a compact state that was not present in CTD2B. The origins of this are not clear given the more extended conformations seen in rxDMD (Figure 2A). In contrast to DMD, where CTD2 was free at both ends, we attached CTD2A to the surface using N-terminal biotinylation, which in some ways mimics the directional membrane attachment from palmitoylation by restricting the conformational space. Previously, we showed that directional attachment of CTD2B to the surface favored polypeptide compaction [15], which may be the origin of this effect in CTD2A.

GluN2A and GluN2B generate different signaling outcomes due in part to differences in protein interactions with the CTDs [23,69]. For GluN2B, these interactions involve liquid-liquid phase separation with PSD-95 and synGAP [25,26,28]. We used the same synGAP construct, which contains only ~12% of the native protein including the coiled-coil (CC) domain, which drives synGAP multimerization, and the PSD-95 binding motif (PBM) [25]. We were able to reproduce the published results with CTD2B but did not see condensates with CTD2A using three different measurements for condensate formation: turbidity, sedimentation with SDS-PAGE, and DIC microscopy. Thus, CTD2A is more likely than CTD2B to self-associate into a colloidal suspension but less likely to participate in LLPS with PSD-95 and synGAP. This could be due to differences in the interaction with PSD-95, which we did not directly confirm. Both CTD2A and CTD2B contain the identical PSD-95 binding motif at their C-termini. However, DMD simulations found that CTD2A showed strong mid-range contacts in this region, which could affect PSD-95 binding. It is also possible that the difference in LLPS arises from the sequence patterning we identified in CTD2B, which would be more favorable for cation–π interactions (Figure 1D). This may help support condensate formation [29,31].

There is a developmental transition in NMDA receptor composition with GluN2A replacing GluN2B at mature synapses, which is driven by gene expression rather than the properties of the CTD [38]. Nonetheless, this transition in isoforms could lead to a difference in LLPS propensity similar to what we observed (Figure 4). Our observation that CTD2A favors the formation of colloidal condensates and eventual solid aggregation would support a liquid to solid phase transition in the postsynapse during development. Indeed, the postsynaptic density of mature synapses, which was one of the early condensates to be identified, had the appearance of a semi-solid in electron micrographs [70,71].

## Figures and Tables

**Figure 1 biomolecules-13-00004-f001:**
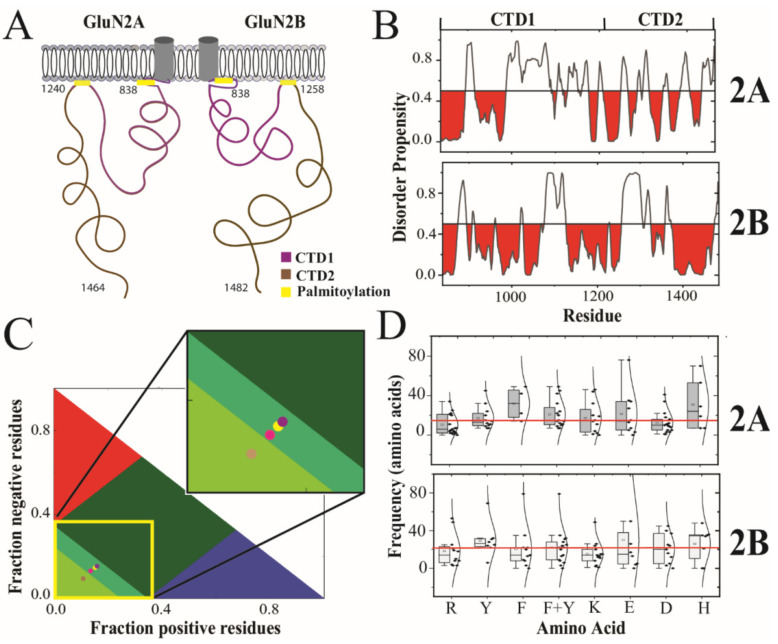
**Prediction and classification of intrinsic disorder in the cytoplasmic domains of the GluN2A and GluN2B.** (**A**) Cartoon schematic of domain organization in the intracellular C-terminal domain (CTD) of the GluN2A and GluN2B subunits of the NMDA receptor. The CTD is connected to the M4 helix within the transmembrane domain [3]. The essential palmitoylation sites (**yellow**) mediate attachment to the membrane [17]. The subdomains demarked by palmitoylation are termed CTD1 (**purple**) and CTD2 (**brown**). (**B**) The disorder propensity from PONDR is plotted for the CTDs of GluN2A and GluN2B with regions predicted to be order-prone (PONDR scores < 0.5) highlighted in **red**. (**C**) Classification of Intrinsically Disordered Ensemble Regions (CIDER) analysis [19] of the CTD subdomains. Colored regions indicate conformational classes of IDPs showing the boundaries for positive polyelectrolytes (**red**), negative polyelectrolytes (**blue**), strong polyampholytes (**dark green**) intermediate polyampholytes (**mint green**) and weak polyampholytes (**pea green**, lower left) [20]. Circles representing the CTD subdomains are placed based on their classification by CIDER analysis. The CTD1 of GluN2A (**yellow circle**) and CTD2 of GluN2A (**magenta circle**) are classified as intermediate polyampholytes. The CTD1 of GluN2B (**purple circle**) is also classified as an intermediate polyampholyte. However, CTD2 of GluN2B (**pink circle**) is classified as a weak polyampholyte. (**D**) The separation between residues within CTD2A and CTD2B are represented by a boxplot with the Gaussian distribution of its recurrence. Shown are the distributions for arginine (R), tyrosine (Y), phenylalanine (F), total aromatics (F + Y), lysine (K), aspartate (D), glutamate (E) and histidine (H). The mean frequency, of all the above-mentioned residues within each isoform, is highlighted with a red line. The standard deviation for the boxplot indicated by black bars.

**Figure 2 biomolecules-13-00004-f002:**
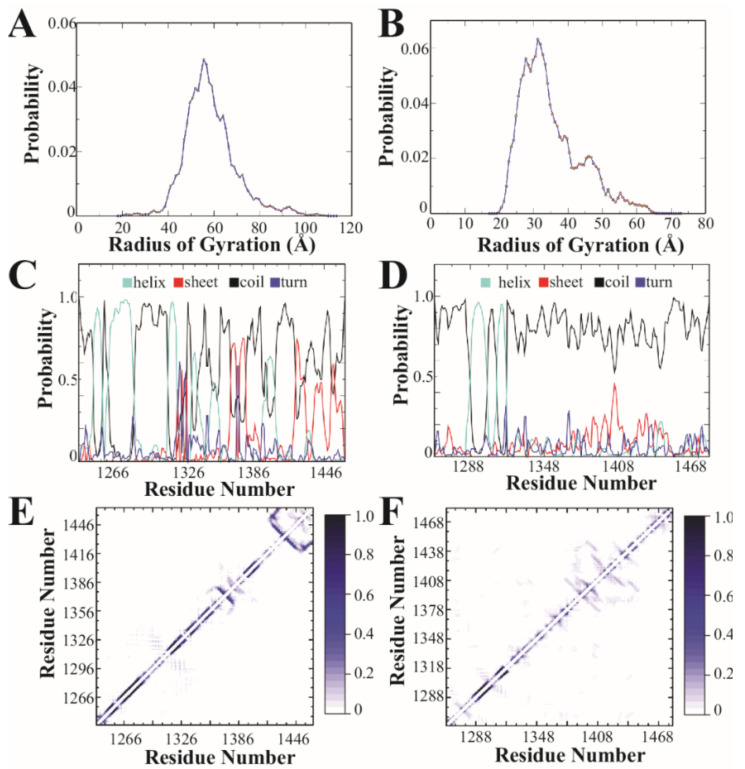
**Discrete molecular dynamics simulation of the CTD2 subdomains from GluN2A and GluN2B.** (**A**) Distribution of the radius of gyration derived from simulations for GluN2A and (**B**) GluN2B. GluN2A favors more extended states. (**C**) The calculated per residue probabilities of different secondary structures based on occupancy observed during simulations for GluN2A and (**D**) GluN2B. Shown are the probability of an individual residue adopting α-helical (light blue), β-sheet (red), random coil (black), and turn (dark blue) conformations. Random coil was the dominant secondary structure for both CTD2s, although GluN2A showed more stable secondary structural elements. (**E**) The pairwise residue-contact frequency maps show the intramolecular interactions observed in simulations of the CTD2 from GluN2A and (**F**) GluN2B. The associated color scale gives the probability of contact between two residues. GluN2A showed stable short-range interactions involved in stabilizing the local, ordered secondary structures. GluN2B showed more long-range contacts.

**Figure 3 biomolecules-13-00004-f003:**
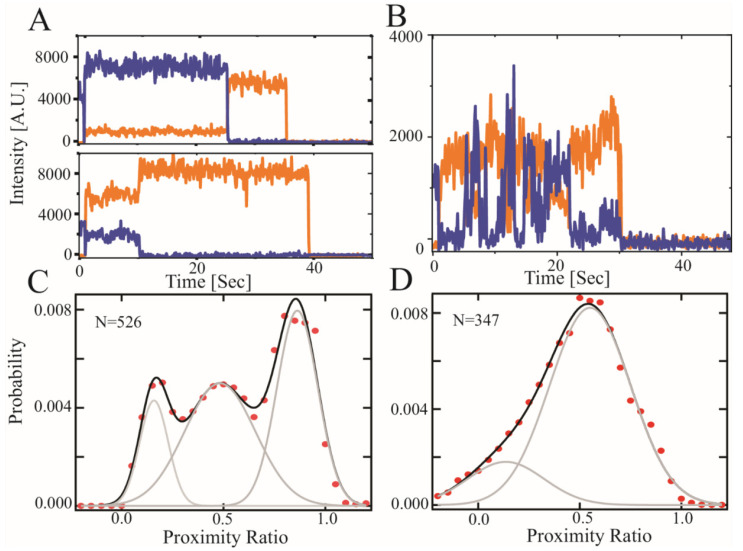
**Single molecule FRET measurements of the CTD2 subdomains from GluN2A and GluN2B.** Representative single molecule intensity time traces for the CTD2 subdomains. (**A**) Representative GluN2A molecules in low and high FRET states. Emission of donor (**orange**) and acceptor (**blue**) fluorophores show stable intensity in GluN2A but vary between molecules within the population. (**B**) Representative GluN2B molecules showing slow timescale, anticorrelated changes in intensity, which is the predominant state as observed previously [15,18,40]. (**C**) Population histogram of raw FRET efficiency (**proximity ratio**) accumulated from each frame captured before photobleaching for the CTD2 subdomain of GluN2A and (**D**) GluN2B. Shown are the experimental data (**red circles**) along with the global fit (**black line**). The number of individual states from global fitting (**grey lines**) differed. GluN2A adopted three states while GluN2B was well fit with a two state model containing wider peaks (Table 2). The number of molecules analyzed is indicated in each panel.

**Figure 4 biomolecules-13-00004-f004:**
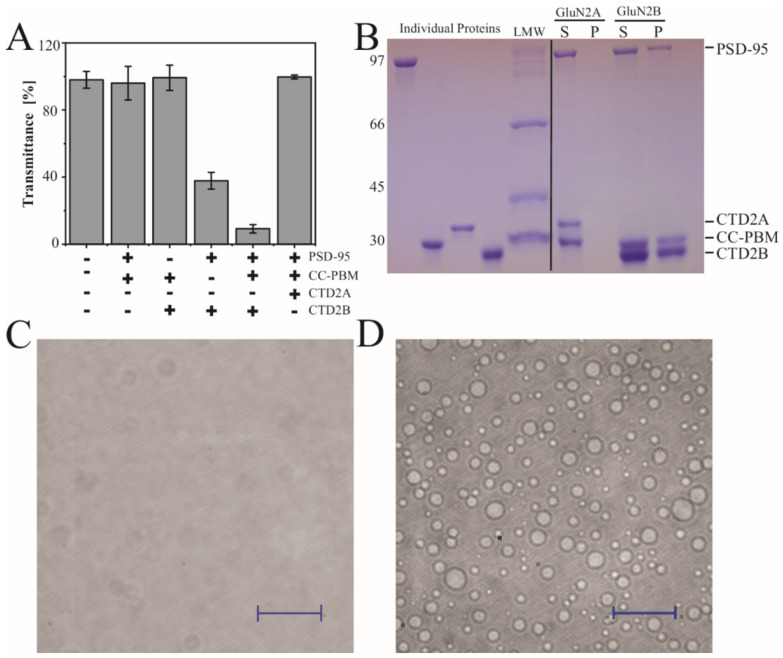
**Condensate formation by the CTD2 subdomains from GluN2A and GluN2B.** (**A**) Measurement of turbidity at 550 nm for the binary and ternary protein mixtures indicated beneath the panel. Samples contained 20 µM of each protein including full-length PSD-95, the CC-PBM fusion from synGAP, and the CTD2A domain from GluN2A or the CTD2B domain from GluN2B. CTD2B shows maximal turbidity while the same concentration of CTD2A remains clear. (**B**) Analysis of protein composition in the condensed phase isolated by sedimentation. Samples were resolved using SDS-PAGE. Left, the individual proteins were run separately followed by the low molecular weight markers (**LMW**). Right, soluble (**S**) and pellet (**P**) fractions from sedimentation of ternary mixtures containing a 1:1:1 ratio of PSD-95, CC-PBM, and CTD2 at 20 µM for CTD2A (**left**) and CTD2B (**right**). The molecular weights are indicated to the left of the gel (in kDa). The identity of each protein band is indicated to the right of the gel (**C**,**D**). Representative images from differential interference contrast (DIC) microscopy of the same ternary protein mixtures used for sedimentation analysis. (**C**) CTD2A does not form droplets, although some scattering is observed at high contrast. (**D**) CTD2B forms droplets with a range of different sizes. The scale bars are 100 µm.

**Table 1 biomolecules-13-00004-t001:** **Sequence analysis of the GluN2A and GluN2B cytoplasmic domains.** The amino acid sequences for the CTDs from GluN2A and GluN2B were analyzed with Classification of Intrinsically Disordered Ensemble Regions (CIDER) [19] The CIDER analysis was performed on the full CTD or the individual subdomains as indicated. The **Sequence** indicates the residue numbers used as boundaries for the analyses of individual subdomains. The **Kappa** value measures the segregation of positive and negative charges within the polypeptide. A kappa value of one indicates a perfect segregation of charge while a value of zero is perfectly mixed. The Fraction of Charged Residues (**FCR**) indicates the ratio of residues containing positive or negative charge to the total number of residues. The Net Charge per Residue (**NCPR**) is the difference between the fraction of positively charged residues and the fraction of negatively charged residues [51]. The **Hydropathy** value reports the mean hydropathy across the indicated polypeptide sequence. The Kyte-Doolittle hydropathy was rescaled from 0 (hydrophilic) and 1 (hydropathy) and then calculated for groups of five residues using a scanning window.

Protein	Sequence	Kappa	FCR	NCPR	Hydropathy
**GluN2A**	838–1464	0.158	0.264	0.018	3.5
**GluN2B**	838–1482	0.183	0.245	0.025	3.6
**CTD1A**	873–1211	0.138	0.286	0.009	3.3
**CTD1B**	874–1212	0.176	0.292	0.009	3.4
**CTD2A**	1243–1462	0.204	0.261	0.009	3.6
**CTD2B**	1250–1482	0.221	0.206	0.026	3.8

**Table 2 biomolecules-13-00004-t002:** **Analysis of the population histograms from smFRET.** The FRET efficiency was calculated from each recorded frame before photobleaching for all molecules containing a single, active donor acceptor pair. These FRET efficiency values were then accumulated into population histograms (Figure 3). These population histograms were fit to a multistate model with an increasing number of Gaussian functions to minimize the fitting statistics. CTD2A required 3 Gaussian states while CTD2B only required 2 Gaussian states. The **Mean** reports the maxima of the Gaussian peak while the **Width** reports the full-width at half height of the Gaussian peak. For these parameters, we report the SEM for three replicate measurements. We also include a simple calculation of the time-averaged distance between the donor and acceptor fluorophores (**<R_DA_>**) in nm for each FRET state based on a self-avoiding random walk (SAW) polymer model [60].

	Mean	Width	<R_DA_>	Mean	Width	<R_DA_>	Mean	Width	<R_DA_>
**CTD2A**	0.21 ± 0.04	0.11 ± 0.01	8.4 ± 0.6	0.46 ± 0.08	0.17 ± 0.01	6.0 ± 0.6	0.85 ± 0.01	0.15 ± 0.02	3.5 ± 0.1
**CTD2B**	0.2 ± 0.01	0.33 ± 0.03	8.6 ± 0.2	0.55 ± 0.05	0.28 ± 0.03	5.3 ± 0.3	NA	NA	NA

## Data Availability

Data from DMD simulations is publically available at https://dlab.clemson.edu/research/NMDA-CTDs accessed on 21 October 2022.

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
