# Peer review of "Different Forms of Disorder in NMDA-Sensitive Glutamate Receptor Cytoplasmic Domains Are Associated with Differences in Condensate Formation"

_biomolecules, 2022, doi:10.3390/biom13010004_

Round 1

Reviewer 1 Report

In the manuscript the authors address the topic of great interest to the scientists working in the field of biochemistry and biophysics of the intrinsically disordered proteins (IDPs) and intrinsically disordered regions (IDRs) involved in the N-methyl-D-aspartate (NMDAR)-type glutamate receptors activities, especially in their dysfunctions. NMDARs are expressed throughout the central nervous system and play the key crucial roles in synaptic function, such as synaptic learning, memory, and plasticity. NMDARs are also involved in the pathophysiology of several CNS disorders. The knowledge describing the structure-function relationships of the intracellular C-terminal domains (CTDs) are highly limited due to their instability and intrinsic disorder. Therefore, the results obtained by the authors are very valuable and significantly expand our knowledge about the properties of these CTDs. The authors compared the CTDs from GluN2A and GluN2B using sequence analysis, discrete molecular dynamics simulations (DMD) and smFRET. The In silico analysis showed that both domains differ in their intrinsic disorder propensities. DMD revealed differences in polypeptide compaction, with GluN2A favouring rather extended states while GluN2B remained globular. The single molecule fluorescence measurements experiments showed that, in contrast to the GluN2B domain, the GluN2A lacked slow dynamics. In order to understand how differences in the form of the intrinsic disorder affect protein interactions,  the authors compared the ability of these two NMDAR isoforms to undergo liquid-liquid phase separation (LLPS), additionally with the SynGAP and PSD-95 proteins. In contrast to CTD2A, the CTD2B isoform readily formed droplets. The manuscript presents novel findings addressing significant biological questions about structural, biochemical and biophysical characteristics of the CTDs of both NMDAR isoforms. This is a very well-written and constructed manuscript describing in details the important structural and physiological issues with the high level of comments and discussions. All necessary techniques and approaches were used in the experiments. In my opinion, the readers of Biomolecules should find the paper worth reading. I have only one comment regarding the LLPS experiments.  Although the results obtained by the authors are rather convincing for me, I would additionally recommend using the fluorescently labelled domains, e.g. by Atto 488 NHS, to confirm clearly the formation of condensates by proteins.

Round 2

Reviewer 2 Report

The authors addressed most of the criticized aspects rigorously. The text reads much more fluently now and the contextual descriptions are clearer.

Apart from some small typos that can be changed during the edition and galley proofs, I have have no concern to say, that this is a proficient study about a very complex topic and the manuscript can be published as is.